# Assisting Personalized Healthcare of Elderly People: Developing a Rule-Based Virtual Caregiver System Using Mobile Chatbot [note 1]

**DOI:** 10.3390/s22103829

**Published:** 2022-05-18

**Authors:** Chisaki Miura, Sinan Chen, Sachio Saiki, Masahide Nakamura, Kiyoshi Yasuda

**Affiliations:** 1Graduate School of System Informatics, Kobe University, 1-1 Rokkodai-cho, Nada, Kobe 657-8501, Japan; cmiura@ws.cs.kobe-u.ac.jp (C.M.); masa-n@cs.kobe-u.ac.jp (M.N.); 2Department of Data & Innovation, Kochi University of Technology, 185 Miyanigutu, Tosayamada-cho, Kami-shi, Kochi 782-8502, Japan; saiki.sachio@kochi-tech.ac.jp; 3RIKEN Center for Advanced Intelligence Project, 1-4-1 Nihonbashi, Chuo-ku, Tokyo 103-0027, Japan; 4Osaka Institute of Technology, 5-16-1 Omiya, Asahi-ku, Osaka 535-8585, Japan; yasukiyo.12@outlook.jp

**Keywords:** virtual caregiver, mind monitoring, personalized healthcare, mobile chatbot

## Abstract

To assist personalized healthcare of elderly people, our interest is to develop a virtual caregiver system that retrieves the expression of mental and physical health states through human–computer interaction in the form of dialogue. The purpose of this paper is to implement and evaluate a virtual caregiver system using mobile chatbot. Unlike the conventional health monitoring approach, our key idea is to integrate a rule-based virtual caregiver system (called “Mind Monitoring” service) with the physical, mental, and social questionnaires into the mobile chat application. The elderly person receives one question from the mobile chatbot per day, and answers it by pushing the optional button or using a speech recognition technique. Furthermore, a novel method is implemented to quantify the answers, generate visual graphs, and send the corresponding summaries or advice to the specific elder. In the experimental evaluation, we applied it to eight elderly subjects and 19 younger subjects within 14 months. As main results, its effects were significantly improved by the proposed method, including the above 80% in the response rate, the accurate reflection of their real lives from the responses, and high usefulness of the feedback messages with software quality requirements and evaluation. We also conducted interviews with subjects for health analysis and improvement.

## 1. Introduction

As an aging population worldwide, the shortage of human caregivers and nursing facilities is becoming a serious problem [1,2,3]. Although the Japanese government is focusing on the shift to in-home care from facility care [4], it takes big burdens on family caregivers, such as 24 h nursing care, one elderly person caring for another, and the elderly living alone. With the big wave of Information and Communication Technology (ICT) and Internet of Things (IoT), study and development for assisting personalized healthcare of elderly people using a machine instead of family caregivers have been gaining popularity in order to reduce families’ burdens [5,6,7,8,9]. In the personalized healthcare of the elderly using machines (i.e., smart healthcare), it often needs two parts of work for elderly people, including situational recognition and care nursing intervention. Studies of situation recognition include in-home fine-grained scenes (i.e., called *context*) recognition [10,11,12]), nonintrusive quality characterization of postural changes [13], and user-defined indoor location sensing [14]. As for care nursing intervention, it is common to use physical care robots [15,16,17,18], chatbots [19,20,21], and virtual agents [22,23,24,25,26]. The core point of these studies is to encourage dialogue-based interaction between machines and elderly people. However, assisting elderly people by merely relying on the progress of study and development is not enough. Specifically, for the case using the technology of nursing intervention for a long-term period, it is still not definite what effect was achieved from the standpoint of the elderly person. That contains not only technical performance, but also its usability and acceptability.

Our interest is to develop a novel virtual caregiver system (called “*mind* monitoring” service) for elderly people. We use the term “mind" to represent an expression from the heart. It includes the expression of mental and physical health states obtained by individual elderly people. Unlike the conventional approach, which detects external situations (e.g., temperature, motion, etc.) using various sensors [27,28], we principally focus on the inner expressions (i.e., thinking and feeling) of elderly people. In previous studies, to realize a smooth care nursing intervention, a speech-based interaction between virtual agents (i.e., MMDAgent [29]) and elderly people on a general computer has been proposed [30,31,32], such as confirming the daily state [33], answering the counseling [34,35,36], and extending to micro service such as watching favorite videos [37,38]. Our key concept is to regard a virtual agent as a caregiver, called a virtual caregiver [39]. We also present a method that allows users to define the appearance of virtual agents [40,41]. The ideal vision of a virtual caregiver is to perform various *person-centric* [42,43] care to improve the Quality of Life (QoL) [44,45,46] of not only elderly people but also family caregivers.

The main contribution of this paper is to implement a rule-based mind monitoring service using a mobile chatbot and apply it to eight elderly subjects (in their 50s to 80s) and 19 younger subjects (in their 20s to 40s) within 14 months to evaluate the effects. In existing approaches with the mind monitoring service, due to maintenance difficulties for the general computers (e.g., OS version update, Internet instability, etc.) at some time, most existing studies have to stop at technical development and short-term experiments. Although some problems relying on the developer using remote desktop tools can be improved, it is still a big challenge to the different network environments. Hence, a method to evaluate the effects of mind monitoring for personalized healthcare of elderly people is desperately needed, especially for evaluating whether elderly people can be accustomed to the form of virtual caregivers. In addition, more useful feedback messages from users need to be collected, and an extensive experiment for many elderly people to improve the monitoring service is also urgent.

The previous version of this paper was published as a conference paper [47]. The most significant change made to this version is the addition of implementation with the mind monitoring service, and experimental evaluation of the collected responses and feedback messages. Unlike the existing approach with the general computer, we advocate implementing the mind monitoring service using the mobile chatbot (i.e., LINE application [48,49,50]). Referring to the definition of health from the World Health Organization (WHO), the light spot of this paper is to integrate a rule-based virtual caregiver system with the physical, mental, and social questionnaires [51,52,53] into the mobile chat application. In the proposed method, the elderly person receives one question from the mobile chatbot per day, and first needs to push the optional button (e.g., have you slept well for the past week? Yes or no.). Then, the mobile chatbot further asks a question to collect detailed information from the last answer. To encourage the usage of the mind monitoring service based on the mobile chatbot, the most workout points in the implementation include developing a method to quantify the answers, generate visual graphs, and send the corresponding summaries or advice to the specific elder.

Based on the implementation of the mind monitoring service, we conducted an actual experimental evaluation of the collected answers, feedback messages, and the mind monitoring service itself with Software Quality Requirements and Evaluation (SQuaRE) [54,55]. As the main result, the response rate for six in eight elderly subjects was above 90%, and the average for 19 younger subjects was around 80%. Meanwhile, the collected response messages reflected the important events in their real lives, such as a lot of stress from COVID-19 due to social activities limited by the curfew. Through the questionnaires with the SQuaRE, most of the subjects were able to reflect on their own state using the implemented service, and interacting with the chatbot was not laborious, nor difficult to answer questions in terms of frequency. Thus, the effects including usability and acceptability of mind monitoring based on the mobile chatbot were significantly improved by the implemented method. The remainder of this paper is organized as follows. In Section 2, we discuss related works on technologies for assisting elderly people, including chatbot technology. We provide a detailed description of the proposed method in Section 3 and the experimental evaluation in Section 4, followed by the conclusions in Section 5.

## 2. Related Work

With the focus on technologies for assisting personalized healthcare of elderly people, exploiting smart care robots [16,56,57,58,59] is a promising way to support the independence of elderly people, as well as to relieve the burden on caregivers and families. However, the use of care robots is not widely spread yet, since deploying and operating a care robot at home is expensive. As one way to solve these challenges, the virtual agent technology [39,40,60,61] has been gathering great attention recently. The virtual agent (VA) is an animated, human-like graphical robot displayed on the screen, and serves as the intuitive user interface of the system through voice interactions in natural language. Since VA only requires a general computer for deployment and operation, the cost is much lower than that of physical robots. Moreover, another advantage is that developers can customize the VA according to the application. MMDAgent is one of the well-known powerful toolkits to implement VAs [29]. Using a three-dimensional movie model of MMD together with built-in speech recognition and synthesis technologies, MMDAgent allows a user to easily develop a custom VA with a spoken dialogue system. In [62], by utilizing the MMDAgent, we have developed a system called virtual care giver (VCG), where the virtual agent named “Mei” provides personalized cares for each elderly person [63]. The VCG supports a wide range of care, such as reminders to check for forgotten items, warnings for forgetting to take a medicine, and playing music videos on YouTube as a value-added service for entertainment [37,38]. Delegating the communication care with the agent, we believe human caregivers can concentrate on *human-centric* tasks that cannot be achieved with the machine. In order to provide appropriate care for each elderly person, it is essential to know what the person is thinking and what the person wants to do. However, internal states such as thoughts, feelings, or emotions cannot be observed from the outside, and cannot be detected by general sensors or the Internet of Things (IoT). Hence, our research group has been developing a new type of sensing technique, called mind sensing (“kokoro” sensing in Japanese) [60,64]. It aims to record the internal mind of a target person that cannot be observed externally by general sensors or IoT. In the mind sensing, firstly, a virtual agent (VA) or a chatbot asks the target elderly person various questions. Then, the person externalizes his/her mind as words by answering the question. Finally, the externalized data are stored as internal states, and utilized for appropriate supports or carers [31,32,35].

The history of chatbot technology dates back to 1950. A mathematician, Alan Turing, proposed the question “Can machines think?” in his paper “Computing Machinery and Intelligence”, which conceptualized chatbots [65]. Then, in 1966, the world’s first chatbot, ELIZA [66], was developed by Joseph, and since then, research on chatbots has developed. Today, the chatbot is a familiar technology for human beings, and it is used in many situations [67,68,69,70,71]. In particular, the use of chatbots in the field of mental health care has attracted a great deal of attention recently, because the chatbot has more potential to make people talk honestly than a real person [72,73,74]. According to the study of Lucas et al. in 2014 [75], it became clear that virtual humans increase the willingness to disclose. More specifically, researchers reported a study in which all participants spoke to a virtual human on a computer screen, but half of the participants were told the virtual human was a computer, and the other half were told it was controlled by an actual human. The results showed participants who believed they were interacting with a computer reported lower fear of self-disclosure, and more willingness to disclose their emotions. These results suggest that automated virtual humans can help overcome a significant barrier to obtaining truthful patient information. From this study, the fact that chatbots also have the potential to make users talk honestly about their feelings and thoughts was found, which led to an increasing usage of chatbots in the mental health field.

One example of developing a chatbot for supporting mental health care was carried out by Kowatsch et al. in 2017 [76]. Since the resources of health professionals are limited and they cannot monitor or support patients in their everyday life, Kowatsch et al. developed a text-based healthcare chatbot (THCB) which can effectively support both patients and health professionals. In the THCB, the main targets are young patients, and they can talk with a chatbot by answering questions regarding their conditions. During the interaction, if the user’s quality of life scores or other critical health states show a negative trend, the chatbot automatically sends notifications to health professionals, so that the professionals can conduct immediate and appropriate interventions. Since the professionals do not have to constantly monitor the status of their patients, it will help to reduce their burden. Other studies include the development of a mental health care chatbot on Discord for people who frequently use their PCs. In [77], the chatbot has a feature that assigns scores to the user’s emotions obtained during interaction with a chatbot, to realize continuous moods tracking of the user. Although the ease of interaction with a chatbot and usefulness of the moods tracking feature was confirmed through a subject experiment, the experiment was small, and further study is needed. Another example is an AI-based virtual health assistant named Vitalk [78]. This is an application for smart phones, and in this application, the chatbot gives advice and health information for users. The chatbot also adapts personalized user’s interests by applying machine learning. The engagement and effectiveness of Vitalk have been evaluated preliminarily in 2020 [79]. Moreover, using a mobile application for ecological momentary assessment (EMA) to collect data on stress and mood in a daily life setting is also presented [80]. The EMA is also called the experience sampling method that asks participants to respond to their thoughts, feelings, and actions and those of their environment in various situations and time streams. Although our approach in this paper is similar to the EMA method, the difference for us is to take older adults who require long-term care at home as the target population. In order to further reduce the burden of family caregivers, a health questionnaire and a chatbot is used to encourage homebound elderly to achieve self-care. The novelty contribution of this paper is that the virtual caregiver regularly asks the elderly appropriate questions from three perspectives: physical, mental, and social. The responses are automatically rated and visualized. This way gives the elderly a clearer picture of their responses over time and encourages better self-care.

## 3. Methodology

### 3.1. Previous Study

In order to provide appropriate care for each elderly person, it is essential to know what the person is thinking and what the person wants to do. However, internal states such as thoughts, feelings, or emotions cannot be observed from the outside, and cannot be detected by general sensors or IoT. An example of comparing the sensor-based and chatbot-based approaches for the elderly healthcare is shown in Figure 1. Hence, our research group has been developing a new type of sensing technique, called mind sensing (“kokoro” sensing in Japanese) [81]. It aims to record the internal mind of a target person that cannot be observed externally by general sensors or IoT. In the mind sensing, firstly, a virtual agent (VA) or a chatbot asks the target elderly person various questions. Then, the person externalizes his/her mind as words by answering the question. Finally, the externalized data are stored as internal states, and utilized for appropriate supports or cares.

### 3.2. Proposed Method

We propose a mind monitoring service in this section. Exploiting the mind sensing service with LINE chatbot, the service tries to externalize the internal state (i.e., mental state) of the target elderly person. The externalized state is recorded for monitoring. Then, feedbacks are sent to the elderly person to make the person reflect and improve his/her living. Figure 2 shows the overall architecture of the proposed mind monitoring service. The service consists of the following three elements: (A1) Interaction with a chatbot using mind sensing service. (A2) Inquiry method specialized for the acquisition of mental state. (A3) Self-care assistance and feedback by monitoring mental state.

#### 3.2.1. A1: Interaction with Chatbot Using Mind Sensing Service

To establish continuous interaction between an elderly person and the proposed service, we exploit the mind sensing service with a LINE chatbot. LINE [82] is a widespread messaging smart phone application in Japan. The reason why we chose the LINE chatbot for the interaction tool is that the chatbot is easier and more reliable to exchange questions and answers, compared to the VA on the PC or email. As explained in the next section, inquiries for acquiring the mental states are somehow technical. Thus, the text message over LINE is clearer than the voice dialogue with the VA. In addition, LINE has an intuitive user interface, as well as text input via voice. In addition, it is already accepted by many elderly people. Hence, we consider that LINE is more suitable than email.

Using the time-based rule of the mind sensing service (see Section 3.1), the LINE chatbot delivers one question per day at the designated time to the elderly person. The time of the delivery can be configured based on the preference. Figure 3 (left) shows a screen shot of the proposed service, where the LINE chatbot asks a question about sleep. As will be explained in the next section, every question simply expects either yes (positive) or no (negative). The elderly person can easily answer the question by tapping one of the choices on the screen. As the elderly person answers the question, an event consisting of timestamps, user ID, question ID, and answer label is notified to our web server. For this, by utilizing the event-based rule of mind sensing service, we command the chatbot to a reply message based on the answer. Within the reply message, the chatbot asks an additional question to investigate more detailed context. Figure 3 (right) shows the case where the user answers “Yes, I’ve slept well.” to the question about the sleep. After receiving the event with a positive answer, the chatbot sends a reply with an additional inquiry to ask about any concerns regarding sleep. The elderly person then inputs detailed information by text. The chatbot finally replies with a stamp (a graphic) as the gratitude for externalizing the details.

#### 3.2.2. A2: Inquiry Method Specialized for Acquisition of Mental State

It is essential for the proposed service to determine what to ask the elderly people and how. Since the primary goal of the service is to support healthy living, we first referred to WHO’s “health” definition [83], capturing the health as a complete state of physical, mental, and social well-being. Hence, we here introduce a framework evaluating the elderlies’ health from the three perspectives physicality, mentality, and sociality. More specifically, the physicality evaluates the physical symptoms that can be seen objectively, such as fatigue, pain, and sleep disorders. Mentality evaluates the subjective feelings such as emotions, moods, and stress. Sociality evaluates the self-evaluations or behaviors of a target person, such as happiness, self-esteem, and social behavior. We then develop questions that acquire the internal states for each of the three perspectives. In our previous study [84], we took questions from GDS-15 [85], PHQ-9 [86], GAD-7 [87], and GHQ60 [88], and implemented 42 questions over the three aspects in the proposed service. However, this approach failed. In the preliminary experiment, there were many complaints from elderly subjects, for instance, “the questions were quite technical and difficult to answer”, “I didn’t understand the meaning of the scale”,“I was tired and gave up for answering too many questions”, etc. Thus, we found that these clinical questionnaires were not suitable to the mind monitoring in their daily living. Hence, we consult a clinical expert to develop more casual questions, so as not to cause too much workload to elderly people. As a result, we have determined seven questions shown in Table 1. To promote easy and quick answers, each question simply expects either yes or no. In the table, “Feature” indicates what to survey by the question. “Category” represents a health category of the framework. In creating questions, we have focused on understanding the person’s conditions approximately. Namely, we narrowed down the seven fundamental features to be monitored within daily living: sleep, health, emotion, memory, psychology, motivation, and socialization. To reduce the burden, we configure the chatbot interaction (A1) to send one question once a day at the designated time. Thus, the seven questions are covered in a week, and the first question is sent again in the next week. Note that each question is asking the “condition of this past week” with respect to the feature.

#### 3.2.3. A3: Self-Care Assistance and Feedback by Monitoring Mental State

Upon receiving every answer from the elderly person, the proposed service stores the answer in our time-series database. Since the seven questions are covered in a week, the proposed service evaluates the collected answers on a weekly basis, and sends feedback based on the result. This section first presents a method of scoring the answers for evaluating the mental states. We then describe a method of generating weekly feedback for promoting spontaneous self-care.

The challenge here is how to quantify (i.e., score) the collected answers. We consulted a clinical expert to consider how a third party (e.g., doctor, caregiver, family) tries to assess the target person’s condition from his/her answer. As a result, we found that there are typically three steps. The first step is to check if the answer is positive or negative, which is the direct objective of the question. The second step is to check how the answer has changed compared to the past. For example, if the answer is negative, we should consider if the answer had been negative for a long time or it has suddenly turned negative. The third step is to understand what the target person said additionally. Considering the above three steps, we propose the following three scoring methods: score answer, score observation, and score sentiment. The three scoring methods evaluate a given answer from different perspectives. Hence, we define the total score $S_{total}$ of the answer to be the weighted sum of them: $S_{total}$ = $w_{1}\cdot S_{answer}$ + $w_{2}\cdot S_{observation}$ + $w_{3}\cdot S_{sentiment}$. Note that, here, $w_{1}+w_{2}+w_{3}$ = 1. Strictly speaking, the weight $w_{i}$ should be adjusted through clinical trials. In this study, however, we simply set $w_{1}=w_{2}=w_{3}=1/3$, since this is the first experiment.

Based on the evaluation result, the proposed service generates and sends feedback containing greeting, summary of the answers, and advice/hints for better living. The purpose of the feedback is to promote the user’s self-reflection and spontaneous mental health care. As explained earlier, the evaluation of the collected answers is conducted on a weekly basis. Thus, the feedback is also generated once a week on a designated day of week. In the proposed service, the weekly feedback is executed as follows: (1) The service selects one question whose answer’s score ($S_{total}$) is the worst in a week. (2) The service generates a feedback message based on the selected answer. To make the message as natural as possible, we structured the feedback message in four paragraphs: (a) greeting, (b) reflection, (c) advice, and (d) conclusion. In (a), the chatbot starts with greeting according to the current season or climate. In (b), the chatbot shows how the user answered the selected question, to get the person to look back him/herself. In (c), the chatbot shows advice or hints for better living with respect to the corresponding question. For this, we refer to the information of “Kenko-Choju Net” [89], which provides knowledge base about health and longevity for Japanese elderly people. In (d), the chatbot concludes the feedback with an encouraging remark. Figure 4 shows an example of the weekly feedback message with the structured four paragraphs. In this feedback, the question about “psychology” was picked up. The chatbot indicates that the user said she had been feeling anxiety before. The chatbot then suggests that she should have her family or friends listen to her anxiety. By receiving such feedback on a weekly basis, users can periodically reflect on their mental states. The service can also promote the user’s self-care consciousness.

### 3.3. Implementation

We have implemented the proposed mind monitoring service using the following technologies:Chatbot interaction: mind sensing service [81] with Java, Eclipse Jersey [90], LINE Mmessaging API [48].Question and message editing, nanagement: Google spreadsheet [91].Automatic rule update: Python, cron [92].Database: MySQL 5.7.27 [93].

To achieve efficient mind monitoring, we have also implemented a web application, called Data Visualizer, which supports users to see the collected time-series data. Data Visualizer is not only for the elderly person him/herself, but also for surrounding people (e.g., family, caregivers, and doctors) who want to monitor the target person. The technologies used for Data Visualizer are as follows.

Development language: Java, JavaScript, HTML, CSS [94].Web service framework: Spring Boot 2.3.0 [95].Template engine: Thymeleaf [96].JavaScript library: jQuery 3.5.1 [97], Chart.js 2.7.2 [98].Web server: Apache Tomcat 9.0.29 [99].

Data Visualizer is mainly designed for smart phones and has the four features.

Reviewing answer logs: The user can reflect the interaction with a chatbot.Weekly score check: The user can check the score of mental states for each week.Monthly score check: The user can check the score of mental states for each month.Mind Monitoring: The user can check the score of mental states for each year.

Figure 5a,b show the top screen of Data Visualizer and Mind Monitoring feature page, respectively. In the graph of Figure 5b, the vertical axis represents the score value and the horizontal axis represents months. The horizontal axis can also be changed to weeks. The scores are averaged over the month for each of the categories: physicality, mentality, and sociality.

## 4. Evaluation

### 4.1. Experimental Setup

To evaluate the proposed mind monitoring service, we conducted an experiment in a real setting. In the experiment, subjects operate the service with their own smart phones for a long term. The operation period was from 1 November 2019 to 31 January 2021, one year and two months (14 months) in total. The demographic characteristics of the participants in the experimental evaluation are shown in Table 2. Eight elderly subjects (in their 50s to 80s) participated in the experiment. In addition to them, we also recruited 19 men and women in their 20s to 40s. The proposed service was originally intended for the elderly generation. Meanwhile, we also wanted to evaluate the effectiveness of the proposed service itself. This experiment was conducted by asking acquaintances of the research group members and others willing to cooperate. During the experiment, if the subject feels any burden, the person can quit answering questions from the chatbot at any time. In addition, we explained to the subjects that the logs of their interactions with the chatbot would be stored in our private database, and would not be disclosed to any third party except the experiment administrators. The experiment was approved by the research ethics committee of Graduate School of System Informatics, Kobe University (No. R01-02). Written informed consent was obtained from subjects for publication of this paper and accompanying images. Based on the data collected in the experiment, we evaluate the proposed mind sensing service from the following viewpoints: (1) responsiveness and continuity; (2) findings from mental state data. (3) Investigation of quality in use.

### 4.2. Results

#### 4.2.1. Responsiveness and Continuity

Our first interest here is to see if elderly people can keep using the proposed service in their daily life. In the experiment, two elderly subjects quit using the service within a few months. As for one elderly person (male in the 70s), it was difficult for him to operate the service, because he was not using the smart phone frequently in his daily life. The other elderly person (male in the 70s) had been using the service for the first three months, but he eventually stopped using it. This was because his asthma worsened and the disease made it difficult for him to answer questions from the chatbot every day. For the remaining 25 subjects, we were able to get them to use the proposed service continuously. We especially show the response rates of elderly subjects for the entire 14 months in Table 3. The response rate is calculated by the ratio of the number of questions answered by the subject to the number of questions sent by the chatbot. From Table 3, we can see that four out of six elderly subjects responded to more than 90% of the questions from a chatbot. For subjects C and E, the overall response rate was low, not because they had stopped using the service, but simply because they responded less frequently. In other words, we could not obtain high frequency of responses from subjects C and E, but we were able to get them to answer the questions periodically. Regarding the other 19 subjects (in their 20s to 40s), the average of the response rates was around 80%. Considering these results, it was confirmed that the responsiveness and continuity of the service were relatively good.

#### 4.2.2. Findings from Mental State Data

Our next interest was to see if the collected mental state data really reflect the minds of elderly people. Through 14 months of the experiment, the service obtained a large amount of time-series data. Using Data Visualizer, we first investigated the time-series data of each subject. Then, we conducted interviews with some elderly subjects to ask what happened by showing the visualized data. Figure 6 and Figure 7 show graphs of the mental state scores of two elderly subjects, subject A (male in 70s) and subject D (female in 70s), in 2020. In the graph, the vertical axis represents the average score value and the horizontal axis represents months. The blue, yellow, and green lines represent the scores of physicality, mentality, and sociality, respectively.

In Figure 6, we can see that subject A’s scores of each perspective are generally positive. This means his condition is relatively stable throughout the year, and his mental health in terms of physical, mental, and social is maintained to some extent. However, we can see that his physicality score dropped sharply in the middle of May. When we asked subject A for the reason, he told us that he had hurt his leg at that time because of too much walking. Afterwards, thanks to treatment and rehabilitation, his leg finally started to get better around July. In addition, we find that his sociality score dropped in early March. This was because the spread of coronavirus (COVID-19) reduced his opportunities to venture out. For a while after that, he stopped exercising at the gym, and his sociality score continued to stagnate. However, he started to go to the gym again around October, and his sociality score started to increase. In Figure 7, the mentality score and the sociality score of subject D are much lower than physicality score. When we asked subject D about her situation in 2020, she told us that her sister had passed away in January and she had been experiencing a terrible sense of loss. This sense of loss continued until around October, and her mentality went through a series of manic depressive cycles. She also had a lifestyle in which her days and nights were reversed. In contrast, her physicality score tended to be relatively positive, but we can also find a sudden decrease in her physicality score around July. In fact, at that time, she was suffering from dizziness caused by otolith detachment. Later, as she learned how to live well with her illness, her physicality score gradually recovered.

Thus, for the data in the months, we confirmed that the time-series mental state data were well reflected by important events in their real living. On the other hand, for the detail every day, further questioning based on the answer collected the exact **reasons** for physicality, mentality, and sociality positively or negatively. For this, we especially focus on the representative further explanations after the negative sociality answered by subjects in the experiment shown in Table 4. From a lot of text messages, although the detailed reasons were different by individual subjects, we can also see that either the older or younger subjects felt much stress from COVID-19, as the social activities were significantly limited by the curfew.

#### 4.2.3. Investigation of Quality in Use

Our final interest is to see if the proposed mind monitoring service is satisfactory for users including not only the elderly but also young people. We believe that there is a need to analyze the experimented service using objective measures. In particular, the evaluation from the user’s viewpoints (e.g., the usability of the subjects) is indispensable. For this, we conducted a “quality evaluation during use” to evaluate the operation and usability of the system during use. More specifically, we evaluated the proposed service using the international software quality standard, SQuaRE (Systems and software engineering Systems and software Quality Requirements and Evaluation, ISO/IEC 25000 Series) [55].

It defines quality requirements, models, and attributes by which stakeholders evaluate the quality of the system and products from various viewpoints. The quality in use is evaluated through actual usage by the users. For this, we created a questionnaire to cover the above characteristics except for environmental risk mitigation. This is because our service does not have any influence on environmental risks. Table 5 shows the questions and results of the questionnaire regarding the above characteristics. In each question, our proposed service is named Mei-chan, which is the display name of the LINE chatbot, so that users can easily understand it. The questionnaire was answered by 18 subjects. Each question was answered by four level scales: strongly agree, agree, disagree, and strongly disagree. Based on the result, we evaluate the proposed service from the viewpoints of effectiveness, efficiency, usefulness, comfort, and flexibility.


**Effectiveness:**


Effectiveness means the degree of accuracy and completeness with which the user achieves the stated goals. Thus, the effectiveness of the proposed service can be assessed by whether the users were able to reflect on their own state and achieve spontaneous mental health care. From Q1, we can see that most of the subjects were able to reflect on their own states. However, Q2 shows that the subjects could not make much efforts to improve their mental states. Thereby, although we were able to confirm the effectiveness of the service to some extent, it was not sufficient.


**Efficiency:**


Efficiency means the degree of the resources used by a user for achieving a particular goal, so the efficiency of the proposed service can be evaluated by the ease of interacting with a chatbot. From Q3 and Q4, it can be seen that interacting with the chatbot was not laborious, nor difficult to answer questions in terms of frequency. Hence, the efficiency of the service was confirmed.


**Usefulness:**


Usefulness means the degree of user satisfaction. Hence, the usefulness can be measured by how much the user finds the service useful. From Q5, more than half of the subjects answered that interacting with the chatbot was useful for reflecting on their own states, while nearly 30% did not. In addition, since it was revealed that some subjects did not want to continue to use the service in Q6, we could not confirm the usefulness of the service very well.


**Comfort:**


Comfort means the degree of user satisfaction with comfort when using the system or software. Thus, the comfort can be evaluated by the ease of using the service including the operability. Since Q7 and Q8 show that the subjects were able to interact with the chatbot and reflect on their own conditions easily, we can confirm the comfort of the proposed service.


**Flexibility:**


Flexibility means the degree to which a product or system can maintain the quality even in a unexpected usage situation. In the proposed service, users are expected to answer one question from a chatbot every day. However, from Q9, we found that there were some cases where the subjects entered their answers at a later time in batches. Although these cases were not originally intended, the fact that there was no problem in using the service suggests that the service maintained a certain degree of flexibility. In addition, in Q10, the subjects said that this kind of flexibility was important.

### 4.3. Discussion

By looking at the changes in the scores for physicality, mentality, and sociality perspective, we were able to analyze the subjects’ mental states in more detail. Since these data are indispensable for understanding the user context, the ability to continuously obtain such information would be the most beneficial aspect of the proposed service. From further observing mental state scores in this experiment, we found two kinds of characteristics in the results. The first characteristic is acute change. In Figure 6 and Figure 7, two subjects had a sudden decrease of their physicality scores due to injury or illness (reasons collected by the subject’s further explanations). In both cases, there was a tendency that the score of physicality suddenly and drastically decreased, and the mentality and sociality scores also decreased along with the decrease of physicality. The second characteristic is chronic stagnation. Subject D felt disorders due to the loss experience from January to October. During this period, the physicality score was basically positive, while the mentality and sociality scores were negative. These results mean that subject D did not have objective symptoms of physical discomfort, but she felt mental discomfort such as lack of motivation. We found that when the scores of mentality and sociality are always stagnant at negative values, even though the physicality score is positive, there may be a possibility that the target person’s invisible disorder is hidden.

Features of the experimented service can be roughly contained into the following: visualization and monitoring of mental states and promotion of spontaneous self-care. Firstly, as for visualization and monitoring of mental state in the experimented service, since we were able to acquire mental state data for more than one year, and to formulate the monitoring process by analyzing the characteristics of scores, we believe this purpose has been achieved. As for the advantages, it is noteworthy that we were able to continuously acquire mental state data during such a long period of operation. We consider that this is because the efficiency and flexibility of the service had a positive impact on the subjects’ continuity. In other words, the fact that it was easy for the subjects to interact with a chatbot and that they were able to use the service whenever they wanted may have promoted the continuity of service use. As for the limitations, promotion of spontaneous mental health care has not been achieved, considering the results of the questionnaire. We thought that reflecting on one’s own condition would lead to self-care, but in fact, although the subjects were able to reflect on their own states, they could not achieve improvements or self-care. In addition, as revealed in the questionnaire, the usefulness of the service was also insufficient. Moreover, there were comments such as “I felt that chatbot’s replies were like fixed form sentences.” or “I think it would be easier to continue if the chatbot could provide more interesting information or unusual topics.” For this, we will devise methods to not make users feel boring, including to change questions or to provide interesting information for users. Meanwhile, we will also plan to focus on two aspects: (1) Analyzing the results of the mind monitoring automatically. (2) Introducing gamification into the mind monitoring in the future.

## 5. Conclusions

In this paper, to assist personalized healthcare for elderly people, we specifically focused on the implementation and evaluation of a virtual caregiver system using a mobile chatbot. The main outcomes include the following: (1) Integrating a rule-based mind monitoring service with the physical, mental, and social questionnaires into the mobile chat application. (2) Developing a method to quantify the answers, generate visual graphs, and send the corresponding summaries or advice to the specific elder. (3) Conducting an experiment to evaluate the effects, including usability and acceptability of mind monitoring based on the mobile chatbot. Through the experiments, the important events in subjects’ real lives could be reflected well by the collected response messages, the usability and acceptability of the proposed mind monitoring are significant, and confirming the advantage that the effect is improved by the implemented method.

Improving personalized healthcare using chatbot technology is a worthy research topic, especially for elderly people, even though this paper has some points that need to be improved. Although the experimental results show that the subjects were able to interact with the chatbot and reflect on their own conditions easily, some subjects could not make much efforts to improve their mental states. We consider that the *individuality* of its effects still needs to be deeply studied as a case study, including personal culture, habits, and life rhythms. Furthermore, integrating chatbot and sensor-based technologies is also an interesting task for us. Unlike the current time-based questionnaires for a static one-week window, it can be developed to event-based using the Internet of Things (IoT) technologies and sensor devices, such as physical sensors (e.g., motion and press) and environmental sensors (e.g., temperatures and humidity). Meanwhile, the retrieved feature values can apply to the Event Condition Action (ECA) rules. Users can define these conditions or generate them automatically by machine learning or deep learning. In this way, it can achieve the dynamic window size of the questionnaire depending on the individual events and users. Furthermore, linking a chatbot and wearable smartwatch to achieve real-time health tracking and timely personalized healthcare questionnaires is also a promising direction in the future.

## Figures and Tables

**Figure 1 sensors-22-03829-f001:**
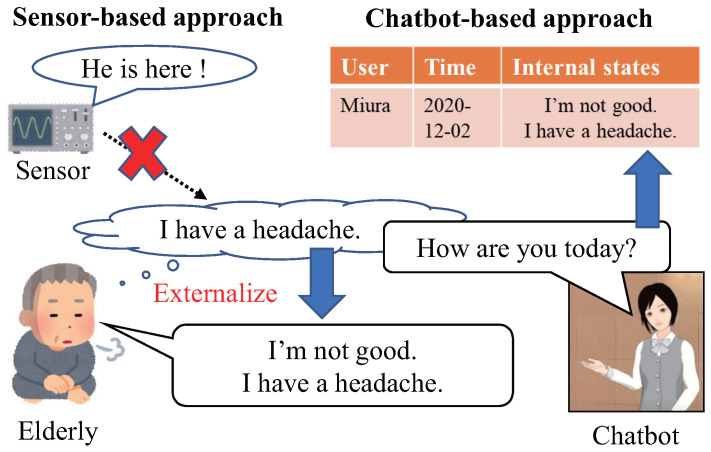
The comparison of the sensor-based and chatbot-based approach for elderly people. Copyright 2009–2018 Nagoya Institute of Technology (MMDAgent Motion “Info of Mei”).

**Figure 2 sensors-22-03829-f002:**
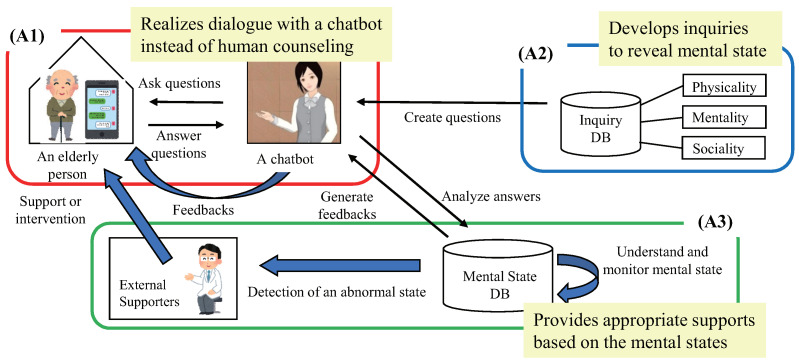
Overall architecture of the proposed rule-based virtual caregiver system using mobile chatbot.

**Figure 3 sensors-22-03829-f003:**
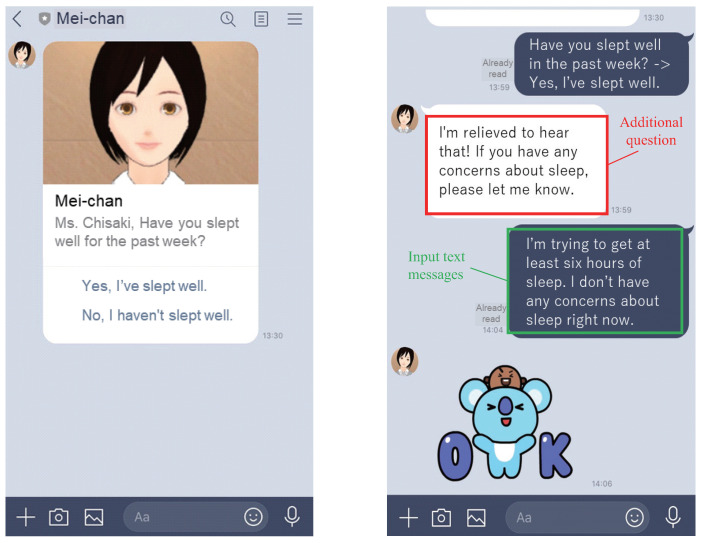
Screenshots of user’s LINE application operated by the proposed service (added notes).

**Figure 4 sensors-22-03829-f004:**
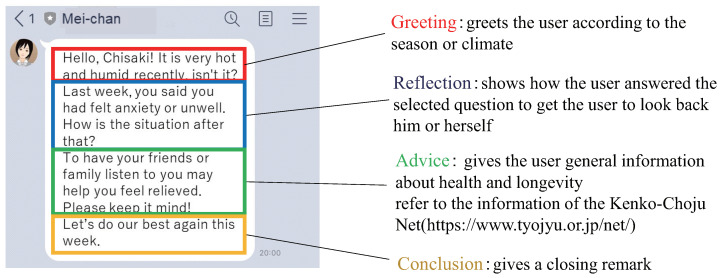
Example of the weekly feedback message with the structured four paragraphs.

**Figure 5 sensors-22-03829-f005:**
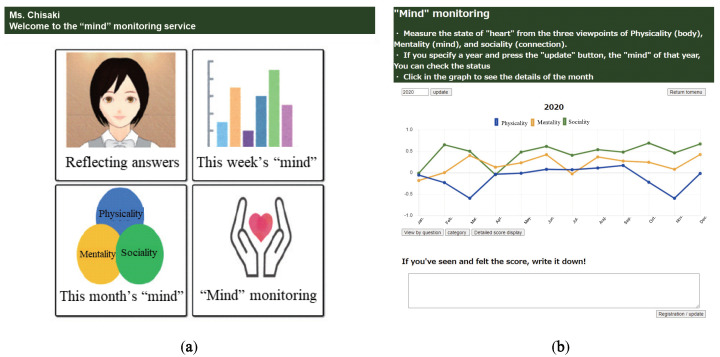
Screenshots of the implemented web application: (**a**) Data Visualizer. (**b**) Mind Monitoring.

**Figure 6 sensors-22-03829-f006:**
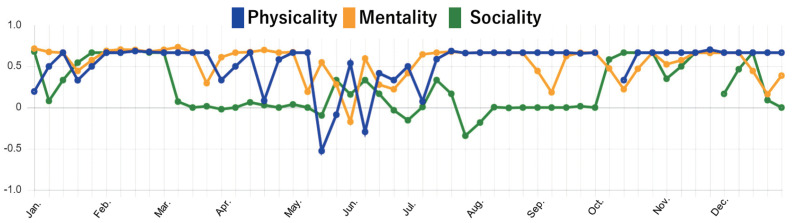
Self−evaluation results of subject A in the period of this experiment.

**Figure 7 sensors-22-03829-f007:**
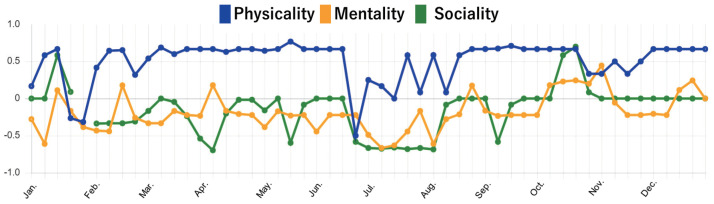
Self−evaluation results of subject D in the period of this experiment.

**Table 1 sensors-22-03829-t001:** Seven questions for capturing the health state from physical, mental, and social well-being.

Question	Survey Item	Category
Have you slept well in the past week?	Sleep	Physicality
Have you felt sick, pain, or tired during the past week?	Health	Physicality
Have you had something fun in the past week?	Emotion	Mentality
Have you felt you could not remember something, or forgotten something in the past week?	Psychology	Mentality
Have you felt anxiety or unwell during the past week?	Psychology	Mentality
Have you felt not motivated or appetite in the past week?	Motivation	Sociality
Have you had many opportunities to go out, to talk and to have hobbies in the past week?	Socialization	Sociality

**Table 2 sensors-22-03829-t002:** Demographic characteristics of the participants in the experimental evaluations.

Demographic Characteristics		Frequency	%
Gender	Male	16	59.3
	Female	11	40.7
Total		27	100
Age	20∼29	12	44.4
	30∼39	4	14.8
	40∼49	2	7.4
	50∼59	2	7.4
	60∼69	1	3.7
	70∼79	5	18.5
	≥80	1	3.7
Total		27	100
Marital status	Married	13	48.1
	Single	14	51.9
Total		27	100

**Table 3 sensors-22-03829-t003:** Total response rate of elderly subjects.

Subject	Age	Gender	Rate
A	70∼79	M	91%
B	60∼69	M	92%
C	80∼89	F	30%
D	70∼79	F	90%
E	70∼79	F	54%
F	50∼59	F	95%

**Table 4 sensors-22-03829-t004:** Representative further explanations after the negative sociality answered by subjects.

Answers in Text Messages to “What Is the Reason for Sociality Negatively?”	Age and Sex of the Subject
I don’t want to go out because of coronavirus.	Female in 80s
I’m bored as I can’t go out much because of the coronavirus.	Female in 50s
I don’t get to meet many people, and my life consists mainly of being at home, which is not that much fun.	Male in 70s
The curfew has made me even less inclined to go out, my body has become even more stiff, and I may not be able to walk when the curfew is ended.	Female in 70s
I’ve been a little stressed out by my self-restrained lifestyle. I miss my normal life.	Female in 40s
I miss seeing my friends. I wonder how long this kind of life will continue...	Female in 20s
I can’t help but get stressed out when I stay at home and work in my room.	Male in 20s
Since I don’t leave the house anymore, I am less aware of dates and days of the week, so I don’t remember the timeline of episodes. I especially can’t remember what happened on the weekend.	Male in 20s

**Table 5 sensors-22-03829-t005:** Results of questionnaire.

Question		Strongly Agree	Agree	Disagree	Strongly Disagree
Q1	Do you think you were able to reflect on your mental state through the interaction with Mei-chan?	27.8%	50.0%	16.7%	5.6%
Q2	Do you think the interaction with Mei-chan helped you to learn about your condition and to improve it?	5.6%	55.6%	33.3%	5.6%
Q3	Do you think the daily interaction with Mei-chan was laborious and difficult?	0%	27.8%	27.8%	44.4%
Q4	Do you think it was difficult to answer Mei-chan’s questions because they were so frequent?	0%	11.1%	27.8%	61.1%
Q5	Do you think the interaction with Mei-chan was useful to reflect on your mental state?	16.7%	50.0%	33.3%	0%
Q6	Do you want to continue to use this service?	33.3%	38.9%	16.7%	11.1%
Q7	Do you think it was easy for you to interact with Mei-chan?	72.2%	22.2%	5.6%	0%
Q8	Do you think you were able to check your mental state easily by interacting with Mei-chan?	33.3%	61.1%	5.6%	0%
Q9	Mei-chan can record the data even if you enter the answers at a later time in batches. Did you do this way of answering?	33.3%	11.1%	11.1%	44.4%
Q10	Do you think the flexibility, for example, the service does not require to respond immediately as in Q9, is important?	61.1%	33.3%	5.6%	0%

## Data Availability

Not applicable.

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
