# Peer review of "Assisting Personalized Healthcare of Elderly People: Developing a Rule-Based Virtual Caregiver System Using Mobile Chatbotâ€"

_sensors, 2022, doi:10.3390/s22103829_

Round 1
Reviewer 1 Report
The authors raise a very interesting and timely topic related to the care of the elderly. This is especially important due to the aging of the population. The importance of this subject is also evidenced by the number of scientific publications related to the aging of the society and various technological solutions supporting or assisting the elderly or the sick. In this case, the authors present a solution based on the increasingly fashionable chatbot model. In their research, they analyzed over a long period of time on two groups of patients. Older and reference groups - younger people. Although the methodology and the period of the study deserve praise, my doubts are still raised by a small group of the elderly surveyed.
The authors refer to other works in the field and comment on them accordingly. This proves a good knowledge of the subject. The unquestionable advantage of the authors' research work is the real implemented system, tested for quite a long period of time. The method presented by the authors, applied in the case of the implemented chatbot, looks promising, despite some comments regarding the size of the research group. After all, it is a really good job and ready for implementation.
However, I have one key point. As I have observed, none of the authors is a doctor, let alone a psychiatrist, and in the discussions and conclusions there are many references and comments regarding the health of the studied patients. I don't think it's legal. This should either be consulted with a doctor in a given specialty, or such arbitrary and categorical conclusions regarding health should not be used in conclusions.
Reviewer 2 Report
In this paper, the authors propose and evaluate a rule-based chatbot for elderly people.
Suggestions and questions (answers can/should be used to improve the paper):
1. At the end of the related work section, the authors should provide the novelty/scientific contributions of the study when comparing with previous ones described.
2. Ecological Momentary Assessment (EMA) mobile applications could be contextualized in the related work section.
3. The subsection "3.4. Discussion" is totally out of context. What is its objective? Future work?
4. Text in lines 307-309 are not results. It presents the objective of the evaluations.
5. Detailed demographic characteristics of the participants in the experimental evaluations should be presented.
6. Consider the sentences "In the subsequent interview, some subjects told us that they used the proposed service by means to vent internal stress. This indicates another possibility that the proposed service can be used to eliminate stresses.". How was this possible? Does the proposed service have the objective to "eliminate stresses"?
7. User experience could be evaluated, for example, by using UEQ (https://www.ueq-online.org/). What was the rationale for using SQuaRE?
8. Study limitations must be acknowledged in the discussion. What could be done differently and/or better?
Specific comments:
- The elderly person receive ...and answer -> receiveS / answerS
- We also presented -> present
- 4. Evaluations -> Evaluation (singular, there is only one, right?)
Round 2
Reviewer 2 Report
The authors answered all my questions, and addressed my concerns.